# Improved Planktonic, Benthic Foraminiferal and Nannofossil Biostratigraphy Aids the Interpretation of the Evolution at Hole U1468A: IODP Expedition 359, the Maldives

Silvia Spezzaferri [1,*] , Jeremy Young [2] , Stephanie Stainbank [1] , Giovanni Coletti [3] and Dick Kroon [4,†]

1 Department of Geosciences, University of Fribourg, Chemin du Musée 6, 1700 Fribourg, Switzerland; stephaniehayman.23@gmail.com
2 Department of Earth Sciences, University College London, Gower Street, London WC1E 6BT, UK; jeremy.young@ucl.ac.uk
3 Department of Earth and Environmental Sciences, Milano-Bicocca University, Piazza della Scienza 4, 20126 Milano, Italy; giovanni.p.m.coletti@gmail.com
4 School of GeoSciences, University of Edinburgh, James Hutton Road, Edinburgh EH9 3FE, UK
* Correspondence: silvia.spezzaferri@unifr.ch; Tel.: +41-26-300-8977
† Deceased author.

**Abstract:** Extended shallow carbonate platform, pelagic, and drift deposits were drilled during International Ocean Discovery Program (IODP) Expedition 359 in the Inner Sea of the Maldives. These sediments yield rich and well-diversified benthic, planktonic foraminiferal and nannofossil assemblages spanning from the early Oligocene to the Recent. We present here the shore-based revised integrated biostratigraphy of these microfossil groups at IODP Hole 359-U1468A together with the paleobathymetric reconstruction. Our data suggests the presence of a late Oligocene carbonate platform, marked by the shallowest water depths of the entire sequence of around 80 m. This carbonate platform sequence occurred from around 29 Ma, the extrapolated minimum age estimate, at least up to 27.5 Ma and possibly up to 25.4 Ma. Up the sequence, similar carbonate production conditions occurred until 22.5 Ma across the Oligocene–Miocene transition, equated at 23.04 Ma, with increased water depths >120 m. Notably, in the time interval approximately from 24 to 21.5 Ma, orbitally induced sapropel layers indicate a change of open to restricted circulation. However, at around 22.5 Ma, pelagic deposition at a distal slope occurred with sedimentation rates of 3 cm/years. This initially occurred in water depths of >350 m but gradually reached deposition in water depths of >500 m, which persisted from 21.12 Ma until approximately the extrapolated age of 12.8 Ma. Sedimentation rates gradually increased to 10.5 cm/1000 years at around 450 m below sea floor, marking the initiation of the drift sequence as identified in seismic lines with an age estimate of 12.8 Ma. The initiation of the drift sequence is also marked by a drastic decrease in the preservation of benthic and planktonic foraminifera from good to very poor at around 12.8 Ma. The drift sequence essentially continued to the present day but was interrupted by two events: the deposition of distinct shallow water benthic shoals and a large hiatus. From 12.8 Ma, a shallowing upward bathymetry is suggested by the occurrence of shallow benthic foraminiferal assemblages that close to around 11.93 Ma reached a maximum water depth of 80 m. This shoal then prograded into the basin and persisted at least until 9.89 Ma. Basin conditions with water depths exceeding 500 m were re-established in the upper part of the sedimentary succession after a hiatus spanning approximately from 9.83 Ma to 2.39 Ma, implying that renewed open ocean conditions occurred in the Pliocene–Pleistocene part of the sedimentary record.

**Keywords:** IODP Expedition 359; Site U1468A; biostratigraphy; benthic; planktonic; foraminifera and calcareous nannofossils age model; timing; paleobathymetry; sedimentary facies evolution

## 1. Introduction

The Maldives carbonate system has great potential to serve as a key area for a better understanding of the effects of an evolving Cenozoic icehouse world in the Indo-Pacific realm (e.g., [1,2]). The Recent and Holocene have been investigated at least in some regions of the archipelago (e.g., [3,4]), with the Cenozoic carbonate sedimentary succession mostly assessed from geophysical data, e.g., seismic sections [1,5–7]. Some of the very few available sedimentary sections from this region were drilled at Ocean Drilling Program (ODP) Sites 714, 715, and 716 from within and adjacent to the Inner Sea. ODP Site 714 consists of a 264 m thick succession of periplatform ooze documenting a mixed record of sea level and bottom-current velocity changes [7,8]. It was continuously cored at a water depth of 2038.3 m and recovered foraminifer-nannofossil chalk from the late Oligocene age [9,10]. ODP Site 715 was cored at a water depth of 2272.8 m and recovered a condensed Pleistocene to Miocene sedimentary succession overlying Eocene shallow-water carbonates [10,11]. Late Miocene to Recent sediments were recovered at ODP Site 716, which was drilled within the Inner Sea [10,12]. The upper Oligocene to Recent sedimentary succession was also documented from the industrial well ARI 1 [6], and sequence boundaries were tied with seismic features [1].

The cores retrieved during the subsequent International Ocean Discovery Program (IODP) Expedition 359 "Maldives, Monsoon, and Sea Level" build upon these early works and provide an extended, nearly continuous and unique sedimentary succession from the Inner Sea, ranging from the early Oligocene to the Recent. The aim of this research is to refine the onboard stratigraphy to frame the water depth evolution of IODP Site 359-U1468, leading to a better understanding of the environmental evolution of the sedimentary units within the improved time framework.

## 2. Geological Setting

The Maldivian archipelago is a pure carbonate environment that has developed since the Eocene [13,14] in the central equatorial Indian Ocean. A lower Paleocene volcanic basement served as a substratum for carbonate sedimentation in the early Eocene to Oligocene. During the early Miocene, carbonate production became restricted to narrow atolls at the respective-most ocean-ward areas, forming the double row of platforms as seen today. During the Miocene, platform margins prograded toward the Inner Sea [15]. Current-related large-scale lobate clinoform bodies were deposited in the region since the mid-late Miocene. The entire sedimentary sequence at Hole U1468A records the transition from carbonate platform to pelagic sedimentation and eventually current-dominated sedimentation leading to the formation of drift deposits [15–17].

## 3. Materials and Methods

All the investigated samples are from sediment cores retrieved during IODP Expedition 359, Hole U1468A at 4°55.98 N, 73°4.28 E [15]. This hole was drilled in the Inner Sea of the Maldives archipelago at a water depth of 521.5 m (Figure 1). Hole U1468A is located in one of the channels connecting the Inner Sea to the Indian Ocean.

The recovered succession is composed of eight sedimentary units [15]. Unit I (0 to 45.67 mbsf) consists of light partially lithified packstone to grainstone with abundant planktonic and benthic foraminifera and other bioclasts (e.g., ostracods, pteropods, molluscs, echinoids, and *Halimeda* fragments). The gradual transition from packstone to grainstone is interpreted as the result of a changing current strength. Unit II (45.67–192.46 mbsf) is dominated by unlithified to partially lithified rudstone to wackestone with abundant larger benthic foraminifera (LBF) and other bioclasts. The unit is layered, with layers characterized by a slight erosional base and fining upward trend. Unit III (192.46–296.4 mbsf) consists of a light gray wackestone succession with intercalations of packstone. Planktonic foraminifera are present, while LBF disappear. Bioclasts are present in the sediments and are often heavily recrystallized.

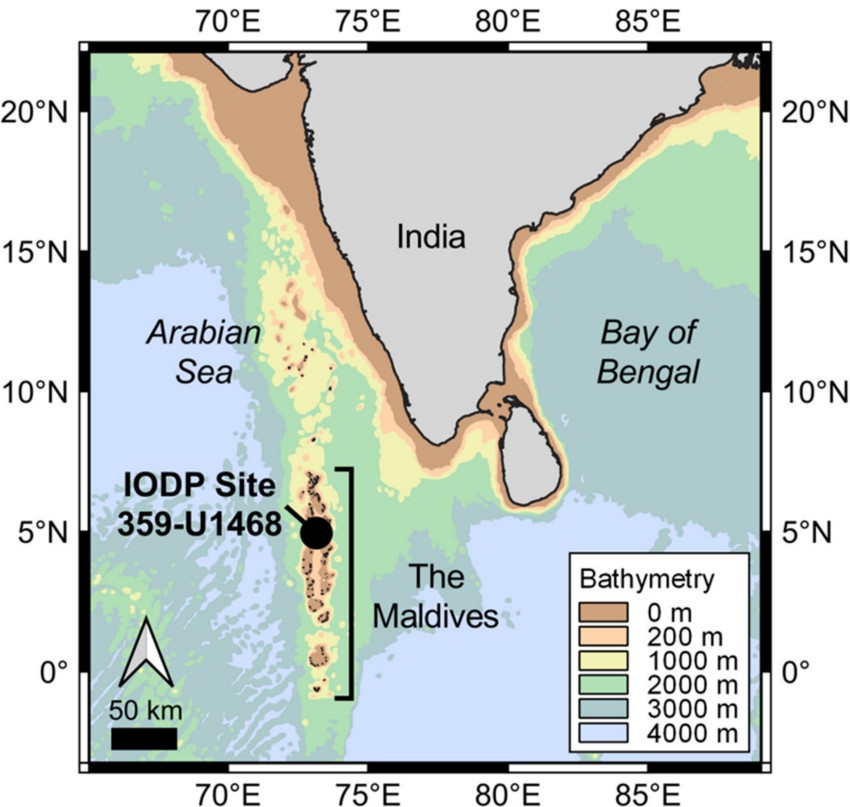

**Figure 1.** Map of the Maldivian archipelago showing the location of IODP Site 359-U1468.

Unit IV (296.4–427.7 mbsf) contains lithified wackestones and packstones with strongly overgrown benthic and planktonic foraminifera. Several ichnotaxa can be observed. Unit V (427.7 to 728.6 mbsf) consists of massive wackestones with alternating light and dark intervals. Planktonic and small benthic foraminifera dominate over other bioclasts. Unit VI (728.6–817.55 mbsf) is composed of wackestones and is characterized by 41 white-dark couplets corresponding to the sapropel interval [15,18]. Planktonic foraminifera and calcareous nannofossils are abundant. Unit VII (817.55 mbsf down to the deepest investigated sample) is a layered wackestone containing abundant LBF. Unit VIII consists of a massive carbonate bank.

Samples for foraminiferal investigations (n = 109), at a ~7.8 m resolution, were dried at room temperature, weighed, and then soaked in water for at least 12 h. In the case of very consolidated chalk samples, they were boiled in diluted $H_2O_2$ to help disaggregate them. Subsequently, they were washed through a 32 μm sieve, dried at room temperature, and investigated for their benthic and planktonic foraminiferal assemblages using a Nikon SMZ18 stereo microscope (Nikon, Egg, Zurich, Switzerland) (Supplementary materials SM1 and SM2).

The planktonic foraminiferal biostratigraphic zonations of [19–21] as modified by [22] were tentatively used for the biostratigraphy. Calibrated ages for bioevents are from [22,23]. The last occurrence (LO) of *Globigerinoides ruber* pink at 0.12 Ma is from [24]. Taxonomic concepts for Neogene and Paleogene taxa mainly follow [25–27] and are synthesized on the Mikrotax website [28].

Minimal and maximal depth distributions of foraminifera are based on literature data (e.g., small benthic foraminifera: [29–39]; LBF: [37,40–49]). Scanning Electron Microscope (SEM) images of foraminifera were obtained at the Department of Geoscience of the University of Fribourg using a SEM FEI XL30 Sirion SFEG (Thermo Fisher Scientific, Reinach TechCenter, Switzerland, Basel).

Light microscopy images were obtained with a digital camera mounted on a Nikon SMZ18 binocular microscope.

Samples for nannofossils (n = 120) at a ~7 m resolution were prepared following the rippled smear slide technique [50] and mounted with Norland Optical Adhesive-NOA61 (Nortland Products Incorporated, New York, NY, USA).

In the case of coarse sand samples, suspension slides were used: a few millimeters cubed of crushed sediment was suspended in water in a 5 mL microcentrifuge tube and allowed to settle for approximately 30 s, and then a few drops of the overlying suspension were pipetted onto a slide. Nannofossil taxonomy follows [51–53] as synthesized on the Nannotax website [54]. The standard zonal scheme of [55] was adopted. However, for age models and intersite calibration, individual events were used according to their reliability and ease of recognition, irrespective of whether they formed part of the standard zonation [15]. The compilations of [56,57] were further used to provide additional information on the reliability, definition, and timing of events. The age vs. depth plots were obtained using TimeScaleCreator v6.3 [58].

## 4. Results (All Biostratigraphic Data Are Reported from Bottom to Top)

*4.1. The Biostratigraphy of Hole U1468A*

4.1.1. Planktonic Foraminifera

A new biostratigraphic investigation of the samples studied on board has been undertaken to integrate the results. In total, 128 planktonic foraminiferal species were identified from IODP Hole 359-U1468A, spanning the Oligocene to Pleistocene, and are listed in the range chart (Supplementary material SM1, Schemes 1 and 2).

In Sample U1468A-110X-CC, 0–7 cm planktonic foraminifera are rare, *Chiloguembelina cubensis* is present, and it contains LBF from the carbonate platform (Supplementary Material SM2). Sample U1468A-109X-CC, 37–42 cm can be attributed to the late Oligocene Zone O4-O5 (29.4 Ma and 27.5 Ma) based on the occurrence of *Paragloborotalia opima*, and planktonic species diversity is generally low (maximum 9). Mixed carbonate platform elements can be observed in this interval [18]. Planktonic foraminiferal assemblages in Samples U1468A-110X-CC, 0–7 cm, and U1468A-108X-CC, 17–22 cm are not age diagnostic.

The interval spanning Zone O7 (25.9 Ma and 22.5 Ma) can be identified from Samples U1468A- 107X-CC, 0–5 cm to U1468A-100X-CC, 33–38 cm. The first occurrences of very small specimens of *Paragloborotalia pseudokugleri* and *Trilobatus primordius* are in Sample U1468A-107X-CC, 0–5 cm (Scheme 1). Zone M1a is not recognised because the marker species *Paragloborotalia kugleri*, which marks the base of Zone M1a, is not present, possibly because of the restricted nature of the basin and/or associated environmental conditions. In [59], the authors provide constraints on the first occurrence of *T. primordius* at 25.6 Ma (Zone O7) on the Ontong Java Plateau (Hole 803D) and Bahamas Bank (Hole 628A). [21,22] place the base of this species in Zone O6 at 26.3 Ma. The absence of *P. kugleri* in Zone M1a is also noted in other restricted basins at this time (e.g., the Parathethys [60]), and this similarity supports the preference of this species for the open sea. This zone is also missing in Hole 714A [61], which was drilled very close to Hole U1468A. Alternatively, a hiatus spanning Zone M1a could also account for the absence of this species, especially considering that in the adjacent and deeper Ocean Drilling Program Holes (e.g., Site 709), this interval is present [9].

Sapropel layers are present in Samples U1468A-104X-CC, 34–39 cm and U1468A-103X-CC, 36–41 cm within Zone O7. Glassy planktonic foraminifera characterize the assemblages and consist of beautifully preserved and very abundant specimens of *P. pseudokugleri*, together with very abundant and large tenuitellids and *Cassigerinella chipolensis*. Sapropel sediments are also found in Samples U1468A-96X-CC, 30–35 cm and U1468A-97X-CC, 44–49 cm within Zone M1b. Planktonic foraminiferal specimens are also very well-preserved (glassy). The weaker alternations of darker and lighter sediment observed up to Sample U1468A-64X-CC [62] are not documented in our samples.

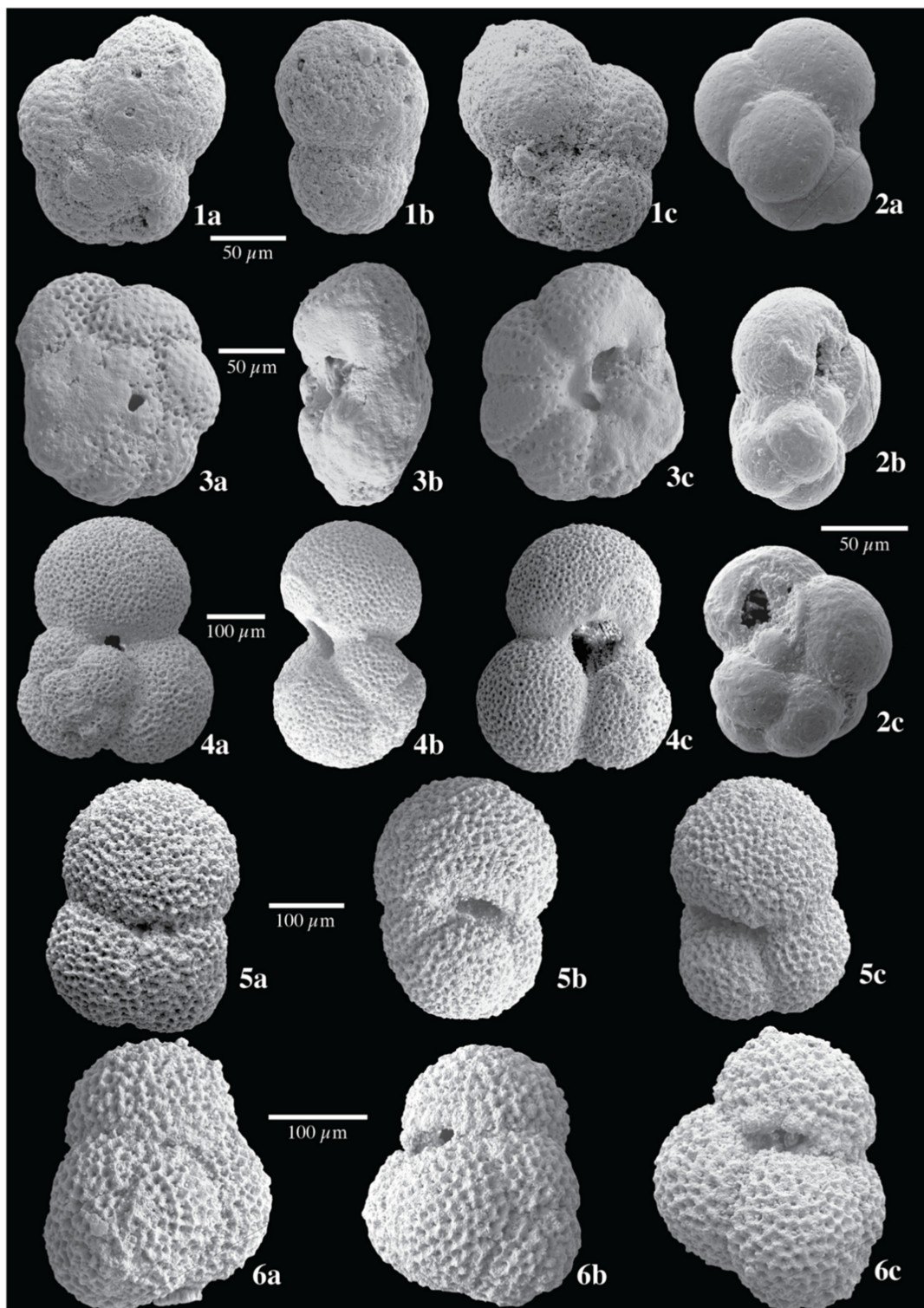

**Scheme 1.** Selected planktonic foraminifera. (**1a–c**). *Paragloborotalia opima* (Bolli), Sample U1468A-109X-CC; (**2a–c**). *Cassigerinella chipolensis* (Cushman and Ponton), Sample U1468A-109X-CC; (**3a–c**). *Paragloborotalia pseudokugleri* (Blow), Sample U1468A-99X-CC; (**4a–c**). *Trilobatus primordius* (Blow and Banner), Sample U1468A-107X-CC; (**5a–c**). *Trilobatus immaturus* (LeRoy), Sample U1468A-99X-CC; (**6a–c**). *Trilobatus* cf *altospiralis* (Spezzaferri), Sample U1468A-99X-CC.

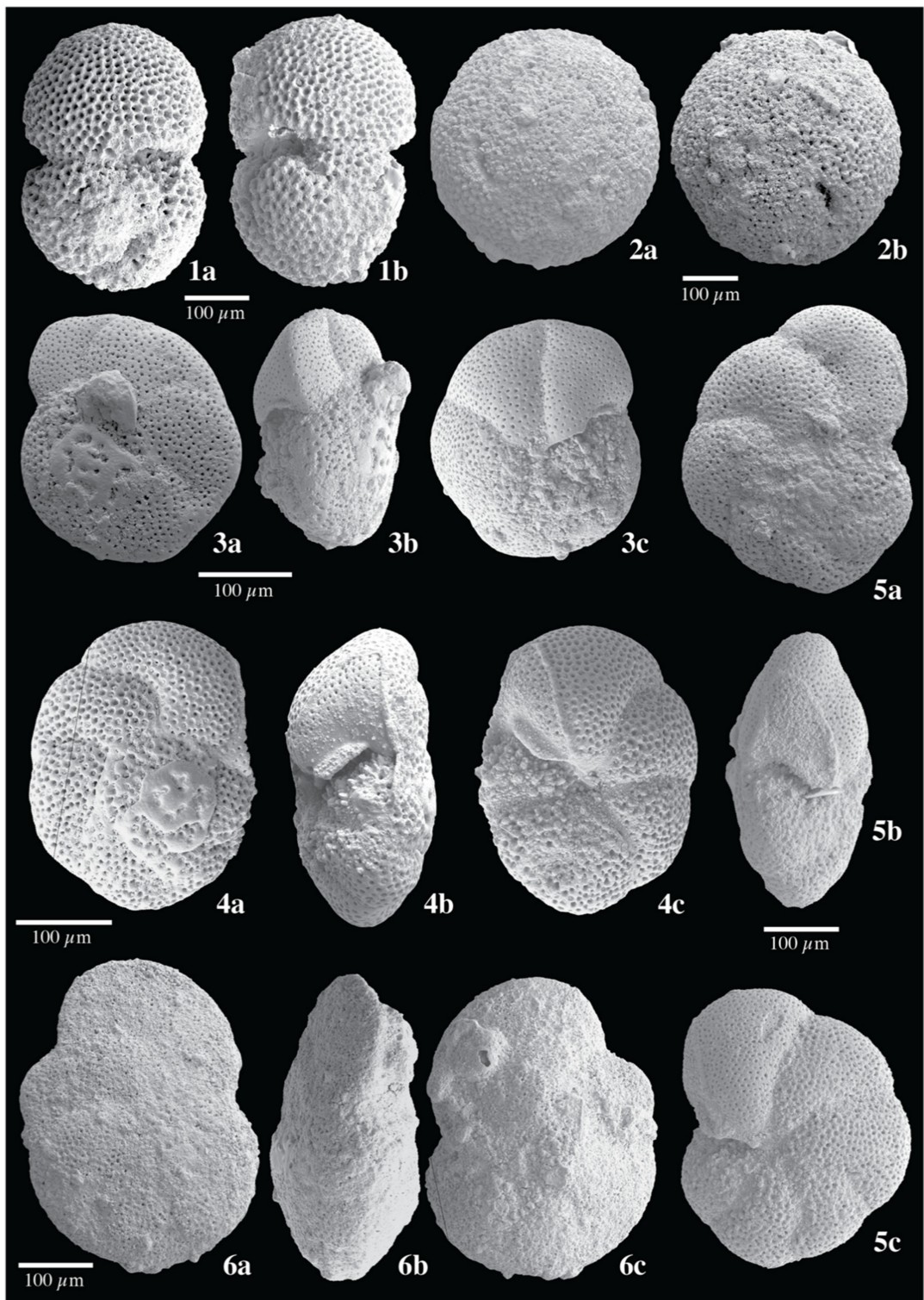

**Scheme 2.** Selected planktonic foraminifera. (**1a,b**). *Praeorbulina transitoria* (Blow), Sample U1468A-76X-CC; (**2a,b**). *Orbulina suturalis* Bronnimann, Sample U1468A-76X-CC; (**3a–c**). *Globorotalia lenguaensis* Bolli, Sample U1468A-72X-CC; (**4a–c**). *Fohsella peripheroronda* (Blow and Banner), Sample U1468A-74X-CC; (**5a–c**). *Fohsella peripheroacuta* (Blow and Banner), Sample U1468A-74X-CC; (**6a–c**). *Fohsella lobata* (Bermudez), Sample U1468A-68X-CC.

The levels in Zone O7 with dominant pelagic deposition, where *C. chipolensis* occurs in the size fraction >250 μm (e.g., Sample U1468A-102X-CC, 38–42 cm) may correlate with

similar intervals described by [9] in the Mascarene Plateau Holes, ODP Leg 115 (e.g., ODP Sites 706 and 709).

*Paragloborotalia kugleri* occurs in the interval from Sample U1468A-99X-CC, 0–5 cm to U1468A-94X-CC, 35–40 cm, together with *Trilobatus subsacculifer*, *T. immaturus*, *T. altospiralis* and *T. trilobus*. These species are typical of Zone M1b. The age estimate of the base is 22.5 Ma. The upper part of Zone M1b is marked by the FO (first occurrence) of *Globigerinoides altiaperturus* in Sample U1468A-96X-CC, 30–35 cm. The FO of *G. altiaperturus* is documented at ODP Hole 516F in the Atlantic Ocean at the base of Zone M2 (=N5) [61,63] at 20.5 Ma [22] and at 19.97 Ma in [23]. However, at the Aquitanian GSSP at Lemme (Italy), the FO of *G. altiaperturus* is documented at 13 m in the sub-chron C6AAr2r, which is in the upper part of Subzone M1b, just below the boundary between subzones M1 and M2 [64,65] at around 21.12 Ma. From this interval up to Sample 67X-CC, 35–40 cm planktonic foraminiferal diversity is higher, ranging from a minimum of 11 to a maximum of 41 species.

The interval spanning Zones M2-M4 (21.12 to 16.39 Ma), cannot be separated because the marker species are missing from Sample U1468A-93X-CC, 0–5 cm to Sample U1468A-88X-CC, 28–34 cm. In particular, the FO of *Globigerinatella* spp. marking the base of Zone M3 at 19.30 Ma is not present or very rarely observed in the Indian Ocean and, in addition, *C. dissimilis* is randomly distributed, and its LO cannot be used to unequivocally indicate the top of Zone M3.

The base of Zone M5a (16.39 Ma [23]) is marked by the FO of *T. sicanus* that is constantly present from Sample U1468A-87X-CC, 0–5 cm with low abundances to samples representing Zone M6. The base of Zone M5b (16.27 Ma) is identified based on the FO of *P. glomerosa* in Sample U1468A-80X-CC, 29–34 cm, and the FO of *Orbulina suturalis* marks the passage to the base of Zone M6 (15.12 Ma), which is identified only in Sample U1468A-76X-CC, 38–43 cm.

The occurrence of *Fohsella peripheroacuta* in Sample U1468A-75X-CC, 30–35 cm allows the identification of Zone M7 (base at 14.06 Ma). The occurrence of *F. praefohsi* from Sample U1468A-73X-CC, 38–43 cm to U1468A-71X-CC, 37–41 cm indicates the presence of Zone M8 (base at 13.77 Ma, Scheme 2).

Zone M9 (13.40 to 11.93 Ma) is identified in the interval from Sample U1468A-70X-CC, 39–44 cm to U1468A-31F-CC, 16–21 cm based on the presence of *F. fohsi*. From the basal interval, which is attributed to Zone M9, up to Sample U1468A-58X-CC, 35–40 cm, planktonic foraminifera indicate a basinal facies characterized by low to high species diversity (4–29 species per sample), although preservation does become very poor. From Sample U1468A-57X-CC, 10–15 cm to the top of the interval attributed to Zone M9, planktonic foraminifera diversity decreases, and the sediment contains higher abundances of benthic shallow water species. This zone is expanded because of increasing sedimentation rates, e.g., 277.6 m of sediments deposited within its duration of 1.62 Ma (Supplementary material SM1).

Planktonic foraminiferal diversity decreases from Sample U1468A-30F-CC, 18–23 cm to U1468A-8H-CC, 15–20 cm (Zone M10 to M11, 11.93 Ma to 11.67 Ma). Zonal assignment is based on the continuous occurrence of *P. mayeri* up to Sample U1468A-8H-CC, 15–20 cm in the absence of *F. fohsi* in Sample U1468A-30F-CC, 18–23 cm. Since *Globoturborotalita nepenthes* is absent in this hole, the transition between Zone M10 and M11 could not be identified. Zone M12 (10.54 to 9.89 Ma), defined by the last occurrence and the first occurrence of *Neogloboquadrina acostaensis*, is not identified in the investigated samples, as *P. mayeri* occurs up to U1468A-8H-CC, 15–20 cm.

The base of Zone M13a (9.89 Ma) is tentatively identified in Sample U1468A-7H-CC, 5–10 cm based on the presence of *Neogloboquadrina* cf. *acostaensis*. Above this sample, a long hiatus occurs, and the biostratigraphic assignment of the sediments overlying the hiatus is very difficult. The zonation of [22] could not be applied, either due to the lack of marker species such as *Globorotalia pseudomiocenica* and *G. miocenica* or because marker species are very rare (e.g., *Dentoglobigerina altispira*). The first apparently reliable datum above the

hiatus is the presence of *Globigerinoidesella fistulosa* in the interval from Sample U1468A-6H-CC, 12–17 cm to Sample U1468A-2H-CC, 7–12 cm. According to the Mikrotax website [28], this species first occurs at the top of the Piacenzian (3.6 Ma) that can be equated to Zone N21 (PL5-PL6) and last occurs at the top of the Gelasian at 1.88 Ma marking the base of Zone PT1. Therefore, the interval characterized by the presence of *G. fistulosa* was initially tentatively attributed to the interval spanning Zones PL5-PL6 (3.47 Ma to 1.88 Ma). The topmost samples of the sedimentary sequence at Hole U1468A do not contain *G. fistulosa* and were tentatively attributed to Zone N22 (PT1). However, very recent studies have highlighted diachroneity in the FO and LO of this species. In particular, it first occurs at 3.155 Ma and last occurs at 1.685 Ma along the North West Shelf of Australia [66]. Additionally, [67] identified its FO between 1.632 Ma and 2.093 Ma and its LO between 2.801 Ma and 3.866 Ma in the Western Pacific, making this datum unreliable.

A comparison with nannofossil data (Figure 2, Supplementary materials SM1 and SM3) indicates that this interpretation may be due to a mixing of specimens from different zones. In particular, according to the nannofossil data, Sample U1468A-4H-CC, 17–22 cm contains large unambiguous *P. lacunosa*, *C. macintyrei*, and *G. lumina* of up to about 4 μm in size, indicating the early NN19/early Calabrian, corresponding to Zone PT1 (base at 1.88 Ma). The flora of Sample U1468A-1H-CC, 15–20 cm indicates the well-preserved Holocene or late Quaternary Zone NN21 corresponding to Zone PT2. *Emiliania huxleyi* first occurs at 0.29 Ma and is abundantly present in this sample. The double interpretation of the biostratigraphy of planktonic foraminifera is given in Figures 2 and 3 and Supplementary materials SM1 and SM3.

### 4.1.2. Benthic Foraminifera

A total of 151 benthic species/taxa were identified from IODP Hole 359-U1468A and are listed in Supplementary Material SM2. Three benthic foraminiferal assemblages can be identified. One low diversity assemblage (the maximum number of species is 6), which contains species from very shallow-water settings (e.g., *Borelis melo* and *Planorbulinella larvata)*, with the published literature indicating living water-depth preferences ranging from 0 to 40 m. This assemblage is here called the "Very shallow" water depth (max 40 m) assemblage (highlighted in yellow and marked with § in Supplementary material SM2).

The second assemblage is characterized by benthic foraminiferal species with water-depth preferences ranging from 0 to 324 m. The depth of 324 m is based on the maximum depth habitat of *Brizalina striatula* collected living and reported in [31]. The maximum number of species representing this assemblage is 22 and is mainly composed of amphisteginids, lepidocyclinids, and *Cymbaloporetta* spp. It is here termed the "Middle-depth" (max 324 m) assemblage (highlighted in light green and with * in Supplementary material SM2). This assemblage may include also some shallow species shed from the adjacent carbonate platform.

The third and well diversified assemblage (maximum number of species representing this assemblage is 28), is characterized by species that have water depth preferences ranging from 420 m down to over 5000 m and contains for example *Cibicides* spp., bolivinids, buliminids, *Trifarina* spp. It is here called the "Deep-water" (>420 m) assemblage (highlighted in pink and marked with <> in Supplementary material SM2).

The base of the stratigraphic succession is identified in Sample U1468A-110X-CC, 0–7 cm containing large benthic foraminifera typical of carbonate platform settings (Middle-depth assemblage): *Operculina*, *Heterostegina*, *Amphistegina mamilla*, and planktonic species are very rare [18] (Supplementary material SM1).

Sample U1468A-109X-CC, 37–42 cm is also characterized by large benthic species, typical of carbonate platform settings, such as *Heterostegina bornensis, Operculina complanata* and *A. mamilla* [18]. These benthic species, together with rare early Oligocene planktonic species (Supplementary material SM1), represent a mixed pelagic and carbonate platform water assemblage.

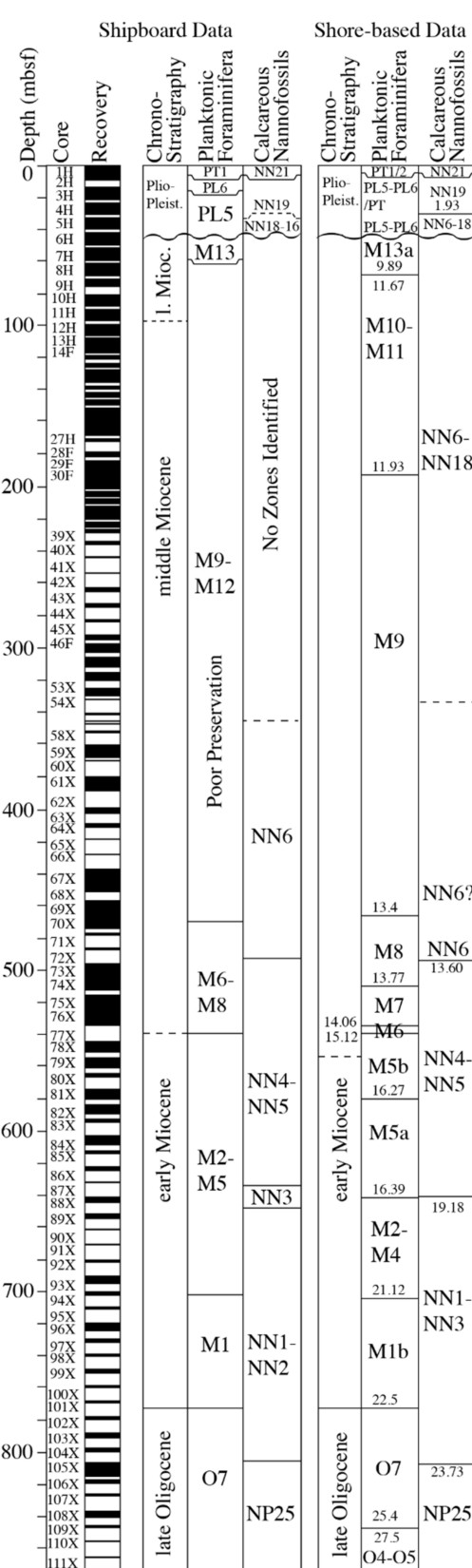

**Figure 2.** Shipboard and shore-based data correlation of planktonic and benthic foraminifera and Nannofossils Zonations are of [19–21], as modified by [22]. Ages (Ma) used in this shore-based study are from [22,23].

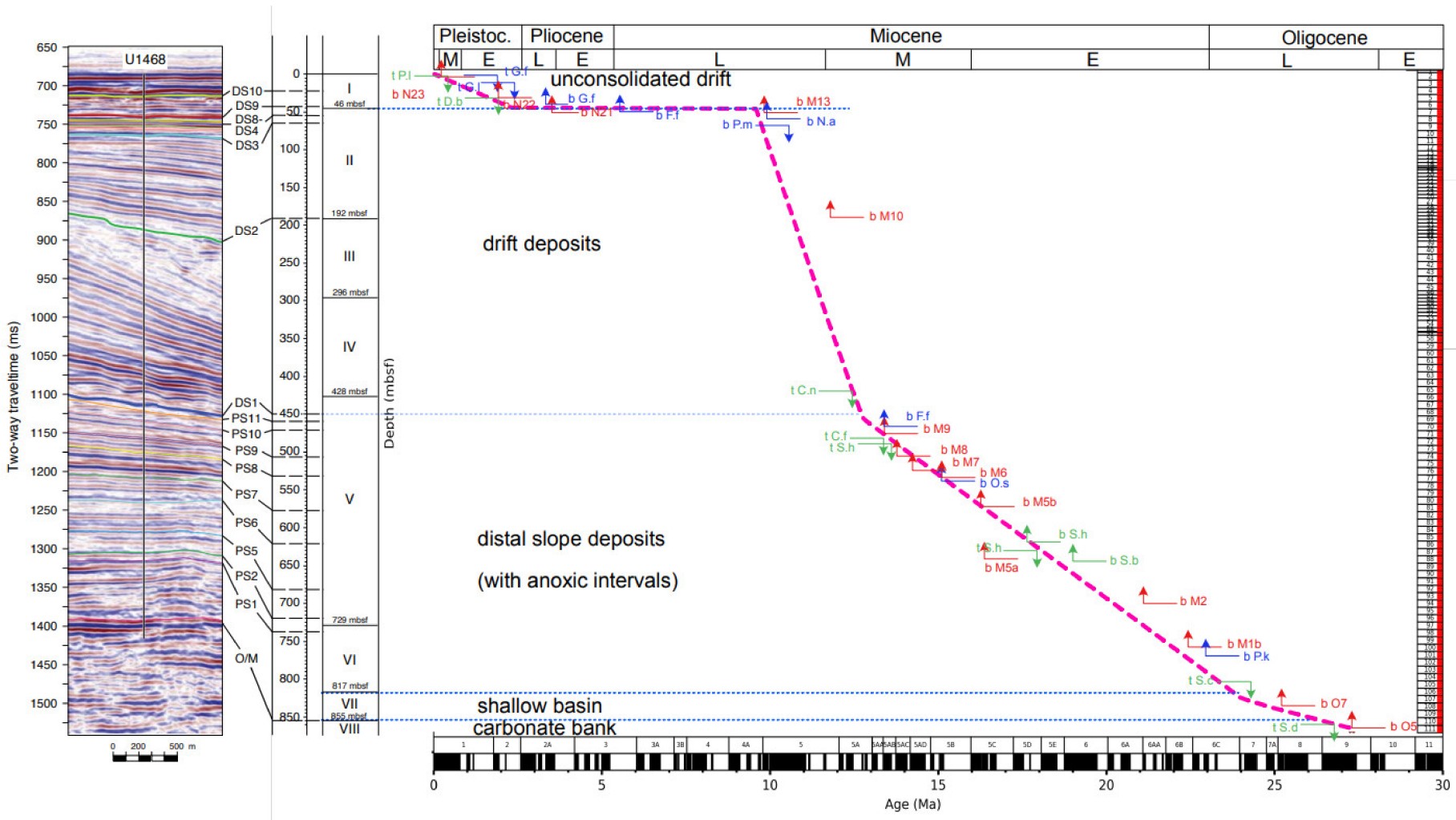

**Figure 3.** Sedimentation rate and seismic profile at IODP Hole 359-U1468A. The age depth/plot is obtained with TimeScaleCreator v6.3 [58]; the seismic profile is from [15]. As only core catchers and a few section samples were used the uncertainty in the position of bioevents is around 9 m. The plot codes are listed in Table 1.

**Table 1.** List of bioevents identified in shipboard and shore-based studies of IODP Hole 359-U1468A together with ages used in this study [22,23]. Plot codes: bE_h = base *Emiliania huxley*; tP_l = top *Pseudohemiliania lacunosa*; tD_b = top *Discoaster brouweri*; tC_n = top *Coronocyclus nitescens*; tC_f = top *Cyclicargolithus floridanus*; tS_h, bS_h = top and base *Sphenolithus heteromorphus*; tS_b, bS_b = top and base *Sphenolithus belemnos*; tS_c = top *Sphenolithus ciperoensis*; tS_d = top *Sphenolithus distentus*; tG_rp = *Globigerinoides ruber* pink; tG_f, bG_f = top and base *Globigerinoidesella fistulos*; tG_l = top *Globorotalia limbate*; bS_d = base *Sphaeroidinella dehiscens*; bN_a = base *Neogloboquadrina acostaensis*; tP_m = top *Paragloborotalia mayeri*; bF_f = base *Fohsella fohsi*; bO_s = base *Orbulina suturalis*; tP_k, bP_k = top and base *Paragloborotalia kugleri*; for the Zones b = base and t = top.

| Name | Plotcodes | Age | Top Depth | Bottom Depth | Data Origin |
|---|---|---|---|---|---|
| FO *Emiliania huxleyi* | bE_h | 0.29 | 3.4 | 8.8 | Shipboard / Postcruise data (this article) |
| LO *Pseudoemiliania lacunosa* | tP_l | 0.43 | 3.4 | 8.8 | Shipboard / Postcruise data (this article) |
| LO *Discoaster brouweri* | tD_b | 1.93 | 31.2 | 40.5 | Shipboard / Postcruise data (this article) |
| LO *Coronocyclus nitescens* | tC_n | 12.45 | 419.2 | 429 | Shipboard / Postcruise data (this article) |
| LCO *Cyclicargolithus floridanus* | tC_f | 13.33 | 473.3 | 497.9 | Shipboard / Postcruise data (this article) |
| LO *Sphenolithus heteromorphus* | tS_h | 13.6 | 486.9 | 497.9 | Shipboard / Postcruise data (this article) |
| FO *Sphenolithus heteromorphus* | bS_h | 17.65 | 624 | 624.7 | Shipboard / Postcruise data (this article) |
| LO *Sphenolithus belemnos* | tS_b | 17.94 | 624.7 | 643.7 | Shipboard / Postcruise data (this article) |
| FO *Sphenolithus belemnos* | bS_b | 19.01 | 644.7 | 653.7 | Shipboard / Postcruise data (this article) |
| LO *Sphenolithus ciperoensis* | tS_c | 24.3 | 806.4 | 807.3 | Shipboard / Postcruise data (this article) |
| LO *Sphenolithus distentus* | tS_d | 26.8 | 846.7 | 880 | Shipboard / Postcruise data (this article) |
| LO *Globigerinoides ruber* pink | tG_rp | 0.12 | 0.05 | 3.35 | Shipboard |
| LO *Globigerinoidesella fistulosa* | tG_f | 1.88 | 3.4 | 8.8 | Shipboard |
| LO *Globorotalia limbata* | tG_l | 2.39 | 8.8 | 21.6 | Shipboard |
| FO *Globigerinoidesella fistulosa* | bG_f | 3.33 | 40.6 | 50.1 | Shipboard |
| FO *Sphaeroidinella dehiscens* | bS_d | 5.53 | 50.1 | 59.4 | Shipboard |
| FO *Neogloboquadrina acostaensis* | bN_a | 9.89 | 59.4 | 69 | Shipboard |
| LO *Paragloborotalia mayeri* | tP_m | 10.54 | 69 | 75.7 | Shipboard |
| FO *Fohsella fohsi* | bF_f | 13.4 | 466.6 | 474.8 | Shipboard |
| FO *Orbulina suturalis* | bO_s | 15.1 | 534.6 | 551.2 | Shipboard |
| LO *Paragloborotalia kugleri* | tP_k | 21.12 | 694.7 | 711.1 | Shipboard |
| FO *Paragloborotalia kugleri* | bP_k | 22.96 | 769.7 | 779.4 | Shipboard |
| base Zone N23 | bN23 | 0.2 | 3.4 | 12.9 | Postcruise data (this article) |
| base Zone PT1 | bPt1 | 1.88 | 31.9 | 41.4 | Postcruise data (this article) |
| base Zone Pl5-6 | bPL5-6 | 3.47 | 50.9 | 60.4 | Postcruise data (this article) |
| base Zone M13 | tM13 | 9.89 | 50.9 | 60.4 | Postcruise data (this article) |
| base Zone M10 | bM10 | 11.93 | 192.4 | 197.1 | Postcruise data (this article) |
| base Zone M9 | bM9 | 13.4 | 476.3 | 486 | Postcruise data (this article) |
| base Zone M8 | bM8 | 13.77 | 505.5 | 515.2 | Postcruise data (this article) |
| base Zone M7 | bM7 | 14.06 | 524.9 | 534.6 | Postcruise data (this article) |
| base Zone M6 | bM6 | 15.12 | 534.6 | 544.3 | Postcruise data (this article) |
| base Zone M5b | bM5b | 16.27 | 573.4 | 583.1 | Postcruise data (this article) |
| base Zone M5a | bM5a | 16.39 | 641.3 | 651 | Postcruise data (this article) |
| base Zone M2 | bM2 | 21.12 | 699.5 | 709.2 | Postcruise data (this article) |
| base Zone M1 | bM1b | 22.5 | 757.7 | 767.4 | Postcruise data (this article) |
| base Zone O7 | bO7 | 25.4 | 835.3 | 845 | Postcruise data (this article) |
| top Zone O5 | bO5 | 27.5 | 865 | 874.7 | Postcruise data (this article) |

Samples U1468A-108X-CC, 17–22 cm and 107X-CC, 0–5 cm also contain larger benthic foraminifera ("Middle depth" assemblage) such as *N. ex. interc. isolepidinoides-sumatrensis*, *Cycloclypeus eidae*, *O.* cf. *heterosteginoides*, *H. bornensis*, and *O. complanata* [18], indicating reworking from the platform at least at the base of this interval. These benthic species occur together with rare specimens of upper Oligocene planktonic species. The "Very shallow" assemblage is very rare throughout this interval, spanning from Zones O4-O5 to the base of Zone O7 (Sample U1468A-107X-CC, 0–5 cm). The components of the "Middle-depth" assemblage decrease in abundance from Sample U1468A-106X-CC, 25–30 cm to Sample U1468A-96X-CC, 30–35 cm, while the benthic species typical of the "Deep-water"

assemblage become prevalent. Species diversity remains low throughout this interval, corresponding to most of Zone M1b. From Sample U1468A-95X-CC, 0–5 cm benthic foraminiferal species diversity increases up to Sample U1468A-58X-CC, 35–40 cm (from the top of Zone M1b to M9 (22.5 Ma to 11.93 Ma), where the "Deep-water" assemblage dominates, although the "Middle-depth" assemblage is also diverse. From Sample U1468A-55X-CC, 20–25 cm up to Sample U1468A-30F-CC, 18–23 cm (Zone M9 to the base of M10-M12, from 13.4 Ma to 11.93 Ma) species diversity progressively decreases to 0. In Sample U1468A-29F-CC, 15–20 cm up to Sample U1468A-6H-CC, 12–17 cm (Zone PL5/PL6), the "Deep-water" assemblage is replaced by the "Middle-depth" assemblage. From Sample U1468A-5H-CC, 58–61 cm to Sample U1468A-1H-CC, 15–20 cm diversity of the "Deep-water" species assemblage increases up to 33, whereas the "Very shallow" and the "Middle-depth" species are present but never abundant nor continuously distributed (Zones PL5-PL6 and PT1).

4.1.3. Calcareous Nannoplankton

A total of 54 nannofossil species have been identified at IODP Hole 359-U1468A. The primary analysis of the nannofossil samples was undertaken on the ship, supplemented by additional observation of selected slides post-cruise. The results are presented in Supplementary Material SM3 and in the age-depth plot (Figure 4).

The basal two samples of the section, Samples U1468A-109X-CC and U1468A-110X-CC, were hard limestones with larger benthic foraminifera. Very rare nannofossils were recovered from Sample U1468A-110X-CC, whilst Sample U1468A-109X-CC yielded only very sparse assemblages of non-age diagnostic nannofossils. Similar to Samples U1468A-110X-CC and U1468A-109X-CC, Samples U1468A-108X-CC to U1468A-106X-CC also yielded relatively sparse assemblages, and no age assignment is made for these (Supplementary material SM3). These cores, related to moderately shallow-water settings (Figure 4), are overlain by basinal sediments of late Oligocene to middle Miocene age (Sample U1468A-105X-CC up to Sample U1468A-64X-CC; 815.52 mbsf–412 mbsf). This extended succession is also characterized by sapropel layers. The most intense sapropels occur in the lower part of this section, Sample U1468A-105X-CC up to Sample U1468A-97X-CC, but weaker alternations of darker and lighter sediment persist through Samples U1468A-96X-CC to U1468A-64X-CC. However, these latter are not observed in the samples used for foraminiferal investigations.

Throughout this interval, nannofossils are abundant to common and well to moderately-well preserved. Generally, the best preserved nannofossils occur in the most intense sapropel intervals. The assemblages in this interval are dominated by *Cyclicargolithus floridanus*, small to medium *Reticulofenestra* spp., *Coccolithus pelagicus*, *Discoaster deflandrei*, *Sphenolithus moriformis*, *Helicosphaera* spp., and *Umbilicosphaera jafari*. *Sphenolithus* species provide a succession of biostratigraphic marker events—top *S. ciperoensis*, base *S. belemnos*, top *S. belemnos*, base *S. heteromorphus*, top *S. heteromorphus*. The last occurrence of *C. floridanus* occurs just above the last occurrence of *S. heteromorphus*, which supports the placement of this event.

These events could all be confidently placed within the core, and they are all well-established and well-dated. In terms of zonation, they allow identification of [55] zones NP25 (Samples U1468A-108X-CC to U1468A-105X-CC, top at 23.73 Ma), NN1–3 (Samples U1468A-104X-CC to U1468A-87X-CC), top at 19.18 Ma, NN3 (Samples U1468A-88X-CC to U1468A-88X-3W, 86 cm) NN4–5 (Samples U1468A-86X-CC to U1468A-73X-CC, 75 cm, top at 13.60 Ma) (Figure 3). Other potential nannofossil markers in this interval did not prove useful: discoasters were generally rather overgrown and no clear specimens of *D. druggii* were observed. There is a distinct succession of *Helicosphaera* species, but the *H. euphratis/H. intermedia* abundance crossover could not be clearly placed, and *H. ampliaperta* was virtually absent.

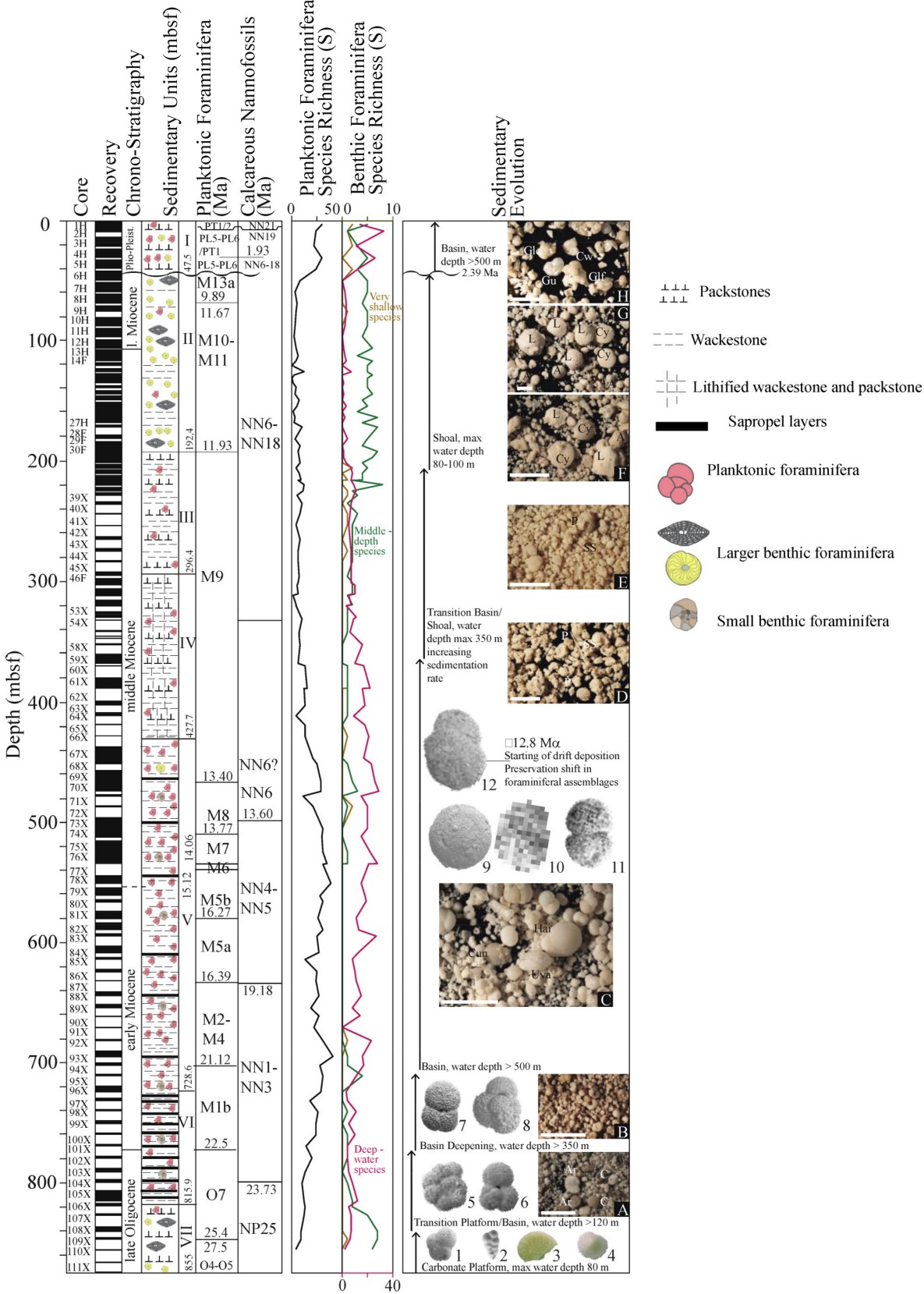

**Figure 4.** Paleobathymetric evolution at Hole U1468A based on benthic and planktonic foraminiferal assemblages. **1**. *Paragloborotalia opima* (Bolli), Sample U1468A-109X-CC. **2**. *Chiloguembelina cubensis*, Sample U1468A-110X-CC. **3**. *Operculina* sp., Sample U1468A-109X-CC. **4**. *Lepidocyclina* sp., Sample U1468A-109X-CC. **5**. *Paragloborotalia kugleri*, (Blow), Sample U1468A-99X-CC. **6**. *Trilobatus primordius* (Blow and Banner), Sample U1468A-107X-CC. **7**. *Trilobatus immaturus* (LeRoy), Sample U1468A-99X-CC. **8**. *Trilobatus* cf *altospiralis* (Spezzaferri), Sample U1468A-99X-CC. **9**. *Orbulina suturalis* Bronnimann, Sample U1468A-76X-CC. **10**. *Fohsella peripheroronda* (Blow and Banner), Sample U1468A-74X-CC. **11**. *Praeorbulina transitoria* (Blow), Sample U1468A-76X-CC. **12**. *Fohsella lobata* (Bermudez), Sample U1468A-68X-CC. (**A**). Overview of Sample 102X-CC: Ar =*Anomalinella rostrata* (Brady), C = *Cibicides* sp., benthic foraminifera. (**B**). Overview of Sample 96X-CC: mixed planktonic foraminifera. (**C**). Overview of Sample 79X-CC: Cun =*Cibicides ungerianus* (D'Orbigny); Uva = *Uvigerina adelinensis* (Palmer and Bermudez), Hai = *Heterolopa haidingeri* (D'Orbigny), benthic foraminifera. (**D,E**). Overview of Samples 40X-CC and 31F-CC, respectively: P = planktonic foraminifera, SS = sponge spicules (**F,G**). Overview of Samples 31F-CC and 16F-CC, respectively Cy =*Cycloclypeus* sp., L = *Lepidocyclina (Nephrolepidina)* sp., A = *Amphistegina* sp. (**H**). Overview of Sample 3H-CC: Gu = *Globorotalia ungulata* (Bermudez), Glf = *Globigerinoidesella fistulosa* (Schubert), planktonic foraminifera; Cw = *Cibicides wuellerstorfi* (Schwager), benthic foraminifera. Scale bars = 1000 μm.

Calcareous nannofossil assemblages from Sample U1468A-63X-CC to U1468A-6H-CC (403.5 mbsf–50 mbsf) become progressively lower in abundance, more poorly preserved, and less diverse up-section. The tops of *Cyclciargolithus floridanus* and *Sphenolithus heteromorphus* distributions both occur between Samples U1468A-54X-CC and U1468A-55X-CC, so the base can confidently be placed in zone NN6. The general assemblage is typical for this interval, as it is dominated by *Reticulofenestra* specimens, including some large (>7 μm) specimens, together with common *Umbilicosphaera*, rare *Sphenolithus abies/moriformis*, and occasional specimens of *Helicosphaera, Pontosphaera, Coccolithus* etc. Discoasters are very rare, overgrown, and never identifiable to species level. No five-rayed discoasters and no *Catinaster* specimens were seen, so an age of U1468A-NN6-7 is likely for these assemblages, but the confidence in this age assignment declines up-section as preservation, abundance, and diversity decrease.

In the upper part of the drift deposit, from Samples U1468A-36F-CC to U1468A-6H-CC (215 mbsf–50 mbsf), nannofossils were very sparse and poorly preserved, often showing very strong overgrowth, and many samples were virtually barren. In these samples, usually only small *Reticulofenestra* spp. and *Umbilicosphaera* were observed, with no sphenoliths or discoasters. The only age inference that can be made for these samples is that they are younger than NN6 due to the absence of *Cyclcargolithus*, and older than NN18 due to the absence of *Gephyrocapsa*. Planktonic foraminifera indicate a late Miocene age for this part of the section; this cannot be confirmed from the nannofossil evidence, but it is entirely compatible with it.

The uppermost 40 m of the section (cores 1–5) are attributed to the Pleistocene Zone NN18-6 (in the modern sense, i.e., including the Gelasian. Three samples have reliably dateable assemblages. Sample U1468A-5H-CC has a poorly preserved assemblage with *Pseudoemiliaina lacunosa*, small *Gephyrocapsa* (<3 μm), and *Discoaster brouweri*. This is clearly late Gelasian in age, Zone CNPL6 of [56]. Sample U1468A-4H-CC has abundant, diverse, and moderately preserved nannofossils including common *Pseudoemiliania lacunosa, Gephyrocapsa* cf. *lumina* (up to about 4.5 μm), and rare *Calcydiscus macintyrei*; this is assignable to the early Calabrian, Zones NN19 and CNPL7 of [56]. Sample U1468A-1H-CC has an abundant, well-preserved Holocene or late Pleistocene assemblage, including *Emiliania huxleyi* that has a FO at 0.29 Ma (identification confirmed by SEM), assignable to zone NN21 of [55] or CNPL11 of [56]. The intervening Samples U1468A-2H-CC and U1468A-3H-CC had low diversity assemblages with few identifiable specimens. The simplest explanation is that this is a continuously deposited sequence with a sedimentation

rate of ca 17 m/Myr, but it is possible that hiatuses or resedimentation occur in the interval of poorly preserved assemblages.

## 5. Discussion

### 5.1. Biostratigraphy and Sedimentation Rates at Hole U1468A

This detailed shore-based investigation of IODP Hole 359-U1468A has allowed the refinement of the onboard planktonic foraminifera biostratigraphy that was initially done under significant time constraints [15]. Additional planktonic foraminiferal Zones (n = 14 in total) are now identified, especially in the Oligocene to Miocene intervals of the sedimentary sequence (Figure 2). Shipboard ages of the identified planktonic foraminiferal bioevents do not differ much from the ages identified here, as reported in Table 1. An important new finding is the identification of Zone O5 based on the presence of very rare planktonic foraminiferal specimens (e.g., *P. opima* and *C. cubensis*) within typical carbonate platform sediments (Figure 4). A late Oligocene age for these carbonate platform sediments is also supported by the LBF assemblage [18]. The early Miocene interval is separated with more detail from Zones M1b to M9.

Calcareous nannofossil biostratigraphy is also refined by adding data from a sedimentary interval previously not zoned (Zones N6/18). The middle Miocene integrated nannofossils and planktonic foraminiferal biostratigraphy is also substantially consistent with the distribution of *Nephrolepidina* species observed and discussed by [18]. On the other hand, the morphology of *Cycloclypeus* species [18], in light of the new data, seems to be a less reliable biostratigraphic index. The topmost part of the sedimentary sequence is characterized by an ambiguous planktonic foraminiferal assemblage that can lead to a double interpretation. We retain the interpretation that is more consistent with using nannofossil data (Figures 2–4; Supplementary materials SM1 and SM3).

The integration of shipboard and shore-based data shows that the bioevents (calcareous nannofossils and planktonic foraminifera), when plotted against depth, define a consistent line of correlation, implying continuous deposition through the 25–13 Ma interval and a sedimentation rate of about 30 m/Myr. No significant discrepancies are observed between the two groups, with the exception of the first occurrence of *Praeorbulina sicana* marking the base of Zone M5a. This discrepancy is possibly due to the poor preservation of the samples that prevented the correct positioning of this bioevent through the sedimentary sequence. These sediments can be interpreted as being deposited in the pelagic environment, possibly along the distal slope of the platform (Figure 4).

The interval from Sample U1468A-68X-CC to Sample U1468A-1H-CC (from 450 mbsf upwards) is interpreted from sedimentology and seismic stratigraphy as being a section through drift deposits, which started at approximately 12.8 Ma and is mainly formed of material derived from the adjacent shallow water carbonate platforms (Figures 3 and 4). This part of the section is expanded and is characterized by high sedimentation rates varying from 30 to 17 m/Myr. The age of the initiation of drift deposition is relatively consistent with the observed change in assemblage composition and poor preservation in foraminiferal assemblages, marking the initiation of shallowing upward water depths accompanied by an increase in grain size at around 12.6/12.8 Ma (Figure 4). A progressive reduction in deposition depth and an increase in grain size occur up-section.

The uppermost 40 m of the section (Sample U1468A-1H-CC to Sample U1468A-5H-CC, down to around 2.39 Ma) is represented in the seismic section by horizontally bedded sediments, being part of the drift deposit and unconformably overlying the hiatus located within Sample U1468A-6H-CC. The sedimentation rate in this interval was around 17 m/Myr (Figure 4).

### 5.2. Paleo-Bathymetric Evolution at Hole U1468A

Benthic foraminiferal associations have well-defined local depth distributions controlled by ecological conditions and different substrata, and as such, their depth ranges may change from one area to another for any given association [68]. However, on a theoretical

basis, the abundance of a given species should be highest where conditions are at their optimum. Nevertheless, individual genera and species have distinctively broad depth ranges. Therefore, it is possible to identify a broad depth division, as indicated for example by [68,69]. To give good estimates of the depth range of benthic species independent from geographical variation, it is essential to consider their general worldwide depth distributions [36]. Although the depth range of each benthic foraminiferal species is not known with precision, indications of these ranges can be inferred from documentation in the literature (e.g., [30–39,41,46,49].

Here, we have selected the minimum and maximum depth of each recognized species (whenever possible) independently from geographic location to further assess the water depth evolution in Hole U1468A. Coupling the distribution of planktonic and benthic foraminifera also allows a better overview of the paleobathymetric history at Hole U1468A (Figure 4).

The abundance of species from the "Middle-depth" water assemblage ([18], Supplementary material SM2) in the lowermost interval indicates that deposition, in an environment characterized by oligotrophic carbonate production, prevailed in the late Oligocene (the interval spanning Zones O4–O5 to the base of Zone O7 from around 29.4 up to 25.9 Ma, Figure 4). This assemblage consists of flat lepidocyclinids associated with *Cycloclypeus* suggesting a water depth from below the wave base to the toe of the slope [44] (Figure 3). This interval has been described as a relatively shallow water deposit overlying the carbonate bank and corresponds to the sedimentary unit VII [15].

The transition platform/basin, possibly driven by a transgression phase, starts at the base of Zone O7 in the late Oligocene at around 25.9 Ma. The basin deepening, expressed in sedimentary unit VI of [15], is indicated by the presence of sub-thermocline planktonic species such as *Globorotaloides* and *Catapsydrax* [70]. These species indicate that the water depth was >120 m, which is approximately the depth of the base of the modern thermocline in the Maldives [71,72]. The sapropel layers present in Zone O7 indicate restricted circulation until almost the middle part of the sedimentary succession, attributed to Zone M1b with an age range from 22.5 to 21.12 Ma [62,73] (Figures 3 and 4).

In the Mediterranean Sea, sapropels are interpreted as being deposited during periods of high productivity and possibly fresh water input (e.g., [74]). The authors of [73] suggest that orbitally driven sapropel formation in the Maldivian region could mainly be related to the waxing and waning of the Antarctic ice sheet. Namely, sea level drops reduced ventilation at the Maldivian sea floor, with only an indirect impact on the surface plankton rather than productivity at the surface. The authors of [62] corroborated this interpretation based on the relatively weak and inconsistent signal given by nannofossils from within the sapropels, with respect to the alternating pelagic sediments. However, the presence of abundant *P. pseudokugleri* may also suggest a slightly higher productivity signal, as suggested by [75], who attributed this species to equatorial upwelling regions with high productivity.

The transition to an open basin with stronger circulation starts at the top of Zone M1b (around 21.12 Ma), as shown by very abundant planktonic foraminifera dwelling from the surface mixed layer to the subthermocline as well as the presence of the benthic species *Cibicides wuellerstorfi*, which indicates water depths > 350 m (e.g., [34]). Basin conditions characterized by pelagic sedimentation prevailed up to the middle lower part of Zone M9 (approximately 12.8 Ma), corresponding to sedimentary unit V of [15] (Figure 4), with water depths >500 m, as indicated by the minimum depth of *Pullenia bulloides* (e.g., [33]).

Increased sedimentation rates (Figure 3), a decrease in species diversity, and a drastic decrease in the preservation of planktonic and benthic foraminifera (Supplementary materials SM1 and SM3) occur in the lower part to the top of Zone M9 at an extrapolated age of 12.8 Ma, according to seismic data, accompanied by a possible shallowing-up trend and increased sedimentation rate (sedimentary Unit V). In this interval, planktonic species co-occur with higher abundances of "Middle-depth" water benthic foraminiferal species

such as amphisteginids, lepidocyclinids, and *Cycloclypeus annulatus*, possibly shed from the adjacent shallow carbonate platforms.

These sedimentary and assemblage changes occur at the same depth (mbsf) in the sedimentary sequence, where Drift Sequence 1 (DS1) was identified in seismic lines and by microfacies analyses [15,17] and interpreted to mark the initiation of the delta drift deposition in the Maldivian Inner Sea (at around 450 mbsf, approximately 12.8 Ma, Figure 3, Supplementary materials SM1 and SM2).

The transition to current-driven deposition and the shallowing up of the basin, here identified at Hole 1468A, roughly coincides also with the drowning of the Kardiva platform and with the onset of the monsoon-driven currents in the Indian Ocean at 13 Ma [15], with a 200 ky difference. This transition is probably driven by a very high sedimentation rate, infilling of the basin coupled with a sea level drop, and the establishment of shallow water shoals conducive for LBF and patchy coral build-ups to thrive. Indeed, a global decrease in the sea level is suggested by global eustatic curves (e.g., [76,77]) in the Serravallian–lower Tortonian. However, water depth at the site of Hole U1468A must have remained >350 m, based on the presence of *C. wuellerstorfi* (e.g., [34]), until the end of Zone M9 (11.93 Ma) in the upper part of the middle Miocene.

The presence of a shallow shoal from the base of the interval spanning foraminiferal Zones M10–13a to calcareous nannofossil Zone NN18, respectively (11.93 to 2.39 Ma), is suggested by the low diversity and scarcity of specimens of planktonic foraminifera and calcareous nannofossils but also by abundant "Middle-depth" benthic taxa (amphisteginids, lepidocyclinids, *Cycloclypeus annulatus*). These forms indicate a maximum water depth of 80–100 m (e.g., [41,44], and they are typically important contributors in delta drift deposits [17].

Basin conditions were re-established in the upper part of the sedimentary succession (corresponding to sedimentary Unit I) after a long hiatus spanning from most of Zone M13 up to the Pleistocene (FO of *E. huxleyi* is observed in Sample 1H-CC and corresponds to 0.29 Ma). The water depth is here inferred based on the co-occurrence of *P. bulloides* and *Ceratobulimina pacifica*, species that are present at water depths usually >500 m (e.g., [33,78]).

The few species and genera belonging to the very shallow assemblage (indicating a maximum water depth of 40 m) are present throughout the sequence at Hole U1468A. These specimens are possibly present due to shedding from the adjacent, very shallow carbonate platforms, which surrounded the site where Hole U1468A was drilled. As these species are never abundant and this assemblage is never dominant, they are more likely to be related to far-field effects rather than to the sedimentary evolution of the site. On the contrary, the large benthic rotaliids that characterize the "Middle-depth" assemblage are often abundant and constitute a sizable portion of the skeletal assemblage of several samples. Therefore, their presence is more relevant for the palaeoenvironmental interpretation, especially concerning the formation of the shoal.

## 6. Conclusions

A shore-based investigation of IODP Hole 359-U1468A was undertaken for planktonic, benthic foraminifera and calcareous nannofossils in order to refine the biostratigraphy of the sediments and to evaluate the water depths of the site for a better understanding of the environmental evolution of the sedimentary units within an improved time framework. The succession of identified shipboard and shore-based bioevents do not differ significantly, although several new biozones have been identified, especially in the late Oligocene to early Miocene part of the sedimentary sequence. Coupling benthic and planktonic foraminiferal assemblages allows a better overview of the timing of water depth changes and the establishment of the drift deposition at Hole U1468A.

Our data indicates that shallow water deposition, characterized by oligotrophic carbonate production, lasted at least to 25.4 Ma, accompanied by basin deepening with water depths >120 m. The sapropel layers present in Zone O7 indicate restricted circulation from around 22.5 to 21.12 Ma.

The transition to an open basin with stronger circulation started around 21.12 Ma, with water depths >350 m. Basin conditions characterized by pelagic sedimentation prevailed up to approximately 12.8 Ma, with water depths >500 m. Drift deposition, marked by Drift Sequence 1, is identified in seismic lines at around 450 mbsf, with an interpolated age of 12.8 Ma; this event is also recorded in a drastic shift in the preservation state of benthic and planktonic foraminifera from good to very poor. From 12.8 Ma, the water depth followed a shallowing upward trend and culminated with the establishment of a shoal close to around 11.93 Ma. The presence of this shallow shoal deposited in the interval spanning 11.93 to 2.39 Ma is suggested by the low diversity and scarcity of specimens of planktonic foraminifera and calcareous nannofossils but also by abundant LBF, indicating a maximum water depth of 80–100 m. Finally, water depths exceeding 500 m were re-established in the upper part of the sedimentary succession from 2.39 Ma up to the Pleistocene, as similarly seen in the present day.

**Supplementary Materials:** The following supporting information can be downloaded at: https://www.mdpi.com/article/10.3390/geosciences12060239/s1, Supplementary material SM1: Distribution of planktonic foraminifera in IODP Hole 359-U1468A, Supplementary material SM2: Distribution of benthic foraminifera in IODP Hole 359-U1468A, Supplementary material SM3: Distribution of calcareous nannofossils in IODP Hole 359-U1468A.

**Author Contributions:** Conceptualization, methodology, validation, investigation, writing—original draft preparation, writing—review and editing, project administration, funding acquisition, S.S. (Silvia Spezzaferri); methodology, investigation, writing—review and editing, validation, J.Y.; writing—original draft preparation, validation, writing—review and editing, S.S. (Stephanie Stainbank); conceptualization, methodology, investigation, writing—original draft preparation, writing—review and editing, D.K.; methodology, validation, writing—review and editing; methodology, writing—original draft preparation, writing—review and editing, G.C. All authors have read and agreed to the published version of the manuscript.

**Funding:** This research was funded by the Swiss National Science Foundation (SNSF) grant Ref. 200021_165852/1 to SSp and the Natural Environment Research Council (NERC) through grant number NERC-NE/N012739/1 to DK.

**Institutional Review Board Statement:** Not applicable.

**Informed Consent Statement:** Not applicable.

**Data Availability Statement:** All data is provided.

**Acknowledgments:** We dedicate this article to our dear colleague and friend Dick Kroon, who sadly and unexpectedly passed away on 24 May 2022. We will miss him immensely. The authors thank the International Ocean Discovery Program (IODP) for providing the samples used in this study. This research was generously funded by the Swiss National Science Foundation (SNSF) grant Ref. 200021_165852/1 awarded to SSp. DK acknowledges funding support from the Natural Environment Research Council (NERC) through grant number NERC-NE/N012739/1. Many thanks to A. Rüggeberg (Fribourg) for some discussion about paleodepth estimates. We acknowledge the constructive comments of the two anonymous reviewers, the academic editors and the MDPI editors, who helped to improve this manuscript.

**Conflicts of Interest:** The authors declare no conflict of interest.

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
