# Peer review of "Improved Planktonic, Benthic Foraminiferal and Nannofossil Biostratigraphy Aids the Interpretation of the Evolution at Hole U1468A: IODP Expedition 359, the Maldives"

_geosciences, doi:10.3390/geosciences12060239_

Round 1

Reviewer 1 Report

This article reports biostratigraphic and paleobathymetric data of the lower Oligocene-Recent sedimentary succession preserved in cores from Hole U1468A, drilled at IODP Site 359 in the Maldives (Indian Ocean).

Results and discussion presented by the AA are interesting and provide a complete and revised biostratigraphic frame as well as a paleobathymetric reconstruction of a key area such as the Maldivian archipelago. This work aims at studying the unpublished material collected during the numerous oceanographic cruises of the IODP program, which is a patrimony for the entire scientific community and needs great attention from researchers.

The MS is quite well organized but also presents some critical parts that needs an improvement. Particularly discussions are not completely supported by results presented: the section dedicated to “benthic foraminifera” and the deriving paleobathymetric reconstruction lack some information. My comments and specific suggestions to the AA are indicated directly on the pdf copy of the MS. The reference list is comprehensive, but 76 different references cited in the text is a high number. I suggest to select only the essential reference papers. I also suggest to reduce some descriptive parts of results by moving the information directly on text-figure 2. I have annotated on the pdf file other suggestions, typing mistakes together with minor comments. In my opinion the manuscript matches the objects of Geosciences and deserves to be published, but before it needs a moderate revision by authors.

Author Response

All words/typos corrections indicated in the manuscript are made (see the mark-ups in the new version.

However, we believe that although the description of the zones for planktonic and calcareous nannofossils may seem redundant in some places, in most cases the zonal attribution is made using assemblages instead of Fist Occurrence and/or Last Occurrences. This is mostly due to the sample resolution of around 7 meters. Placing the FO and LO in Figure 2 without an explanation would result in a loss of important information. Therefore, we would like to keep the description although we realize it is long.

Figure 4 has been modified by adding the curves of the planktonic and benthic foraminifera species diversity and the lithology of the Units.

The references are a lot, we agree with the reviewer, but we believe that they are needed to justify our work, our interpretations and to give credit to who has previously worked on the topic.

It is not possible to add any abundance curve of species as the investigation done is qualitative and not quantitative. However, species abundance curves have been added to Figure 4. Also, it is not possible to give a precise paleobathymetry as variations are clear but cannot be quantified numerically (that can be done only with transfer equations, which was not the purpose of our work), this is why in Figure 4 we give an estimation of paleodepth with > and < than…… xxx meters in certain intervals.

We are aware that the “middle-Depth” assemblage may contain species shed from the platform, and we mention this also in the Discussion. However, a sentence has been added also in the description of the assemblages.

Reviewer 2 Report

The manuscript by Spezzaferri et al. presents an integrated biostratigraphic analysis for IODP Hole U1468A from the Maldives. In addition to biostratigraphy, the authors also conduct a paleobathymetric analysis and infer the ancient depositional history of the area. My below comments are mainly editorial in nature, and are to better clarify the science being presented. Other than these few comments, I see no major issues with the manuscript and look forward to its publication. I commend the authors for such an integrated and detailed biostratigraphic and paleoenvironmental analysis, as such studies are time-intensive but vastly important to the fields of paleontology and sedimentology.

Line 16: ‘Fauna’ specifically refers to animals, thus foraminifera are not fauna. Use ‘assemblages’ or similar instead.

Line 25: Missing a ‘)’

Line 61: Tied with seismic features to what? Site 716? Clarify.

Line 91: Missing a ) at the end of the sentence with Halimeda.

Line 93: First time LBF is used; write out fully.

Line 100: Extra period after 427.7

Line 103: Does sapropel need to be upper case here?

Line 107: Were samples dried prior to weighing?

Lines 114-115: Note that King et al. (2020) provided updated ages through the Neogene (mainly for the Miocene) to the Wade et al. (2011) zonation. This is up to the authors and fine as the zones stand in the manuscript, but if using the Wade et al. (2011) ages, why not also use their revised Cenozoic zones, that incorporate the zonations of Blow (1969, 1979) and Berggren et al. (1995)?

Line 117: A more appropriate reference for pforams @ mikrotax is: Huber, B.T.; Petrizzo, M.R.; Young, J.R.; Falzoni, F.; Gilardoni, S.E.; Bown, P.R.; Wade, B.S. Pforams@ microtax. Micropaleontology 2016, 62, 429–438.

Lines 107-117: Would be good to include the average sample resolution (either in meters or age), same for the benthic foraminifera and calcareous nannoplankton samples.

Line 146 and throughout: Check for double spaces between words and sentences.

Line 207-208: 4 species doesn’t seem like high diversity compared to older intervals with 11 to 41 species. Instead, perhaps modify this sentence to read ‘basinal facies characterized by low to high species diversity’. In line 264-265, benthic foram diversity (n=6) is characterized as ‘low diversity’. I suggest the authors determine what is low, medium, high diversity and standardize language around diversity using their definition (no need to define this in the manuscript, just needs a standardized descriptor).

This is totally up to the authors, but it might be informative to the readers to include a simple plot of diversity (species richness) through the section, perhaps on Figure 4, to better visualize how diversity changes and corresponds to changing facies.

Line 217 and throughout: There are places where Hole is uppercase, and others (line 217) where hole is lowercase. Either is fine, just be consistent.

Lines 227-244: Also note that other authors have identified diachroneity in the first occurrence of G. fistulosa (e.g., Groeneveld et al., 2021 ‘Dating the Northwest Shelf of Australia since the Pliocene’; Lam et al., 2022 ‘Diachroneity rules the mid-latitudes’), which may be further complicating this matter.

Line 268 and in Supplemental Files: I caution the authors on the use of colors in Excel tables. Often, these features are good for the short-term, but over the long-term, as programs are updated and some features become obsolete, any feature of an Excel table that cannot be captured with a simple text file may be lost in the future. This has happened to older Excel files when I have tried to open them in an updated version of Excel. Instead of using colors in supplemental Excel files, you may instead think of using asterisks or some other type of numerical or alphabetical denotation to best preserve information in such tables. Alternatively, a benthic foram diversity plot could be included on figures in the manuscript (again, this is totally up to the authors).

On the ‘Calcareous nannofossil’ tab of the Supplemental File, there is no explanation as to what the colors mean. If the colors are kept on this table, please include such a description.

Figure 3: Some of the text on this figure (ie, the right y-axis and letters/numbers used to denote events) is quite hard to read. In addition, there appears to be two fonts used on the graph. Can text be made larger, where possible? There should be an expanded figure caption to indicate what e.g., tG_l means and what it relates to. There also appears to be a box around part of the figure, and Jeremy’s name and the date at the top right as well as the small text at the bottom right can probably be deleted. If possible, it may be best this figure is presented in the journal in landscape orientation. Other than these issues, this is a very good and informative figure!

 Line 352: Check spelling of ‘Cyclicargolithus’

Table 1: Species’ names need to be italicized. Also, what is meant by the group column, and what do the Plotcode letters refer to? There are also asterisk on some zones, and § on some nanno names. These should be defined in the table as footers or however the journal formats such information. I’m also not certain what is meant by young and old age? Does this correspond to top and bottom age (depth)? A more impactful column here, rather than group and plotcode, would be midpoint depth and midpoint age. In this way, other folks who use this biostratigraphic data for future studies can estimate depth and age error more quickly from your dataset, and which allows a more honest assessment of the data by including such age and depth errors.

Line 424 and throughout: Myr should be capitalized, there are some places where it is and other places it is not.

Figure 4: Some of the boundary ages and sedimentary unit depths are difficult to read; could these be turned vertically? Are some of the sedimentary unit boundaries supposed to be red? If so, include why in the figure caption.

Author Response

Thanks a lot to the reviewer!
